# Distribution of the Indicator of the Appropriate Admission of Patients with Circulatory System Diseases to County Hospitals in Rural China: A Cross-Sectional Study

**DOI:** 10.3390/ijerph16091621

**Published:** 2019-05-09

**Authors:** Qing Ye, Yan Zhang, Hong-xia Gao, Ying-chun Chen, Hao-miao Li, Hui Zhang, Xiao-mei Hu, Shi-han Lei, Di Jiang

**Affiliations:** 1School of Medicine and Health Management, Tongji Medical College, Huazhong University of Science and Technology, Wuhan 430030, China; yeqing180522@163.com (Q.Y.); yanzhang@hust.edu.cn (Y.Z.); chenyingchun@hust.edu.cn (Y.-c.C.); lihaomiao@hust.edu.cn (H.-m.L.); zhanghui@hust.edu.cn (H.Z.); huxiaomei@hust.edu.cn (X.-m.H.); leishihan@hust.edu.cn (S.-h.L.); jiangdi@hust.edu.cn (D.J.); 2Hubei Provincial Department of Education, Key Research Institute of Humanities & Social Sciences, Research Centre for Rural Health Service, Wuhan 430030, China

**Keywords:** inappropriate admission, circulatory system diseases, indicator distribution, county hospital, rural China

## Abstract

**Background:** The inappropriate admission of patients with circulatory system diseases (CSDs) have contributed to the rapid increase in hospitalisation rates in China. The purpose of this study is to identify the key indicators of appropriate admission and their distribution by analysing CSD cases. **Methods:** A total of 794 records of inpatient CSD cases were collected from county hospitals in five counties in midwestern rural China through stratified random sampling and evaluated by using the Rural Appropriateness Evaluation Protocol (RAEP). RAEP has two parts: Indicator A, which represents requirement for services, and Indicator B, which represents diseases severity. Indicator distribution was analysed through frequency analysis. A three-level logistic regression model was used to examine the sociodemographic determinants of the positive indicators of appropriate CDSs admissions. **Result:** The inappropriate admission rate of CSDs was 33.4% and varied between counties. A2 (Varying dosage/drug under supervision, 58.22%), A8 (Stopping/continuing oxygen inhalation, 38.19%), A7 (Electrocardiogram per 2 hours, 34.22%), A3 (Calculation of intake and output volume, 31.19%) and B14 (Abnormal blood condition, 27.98%) were the top five positive indicators of CSDs. Indicator A (requirements for services) was more active than Indicator B (disease severity). The limitation of the role of Indicator B over time may be attributed to the different policies and environments of rural China and stimulated the increase in inappropriate admission rates. The results of three-level logistic regression suggested that the influence of gender, year, region and disease type on positive indicators should receive increased attention in the evaluation of CSDs admissions. **Conclusion:** This study found that A2, A8, A7, A3 and B14 were the key indicators and were helpful to determine the appropriate admission of CSDs in rural China. Managers may focus on these indicators, particularly the use of indicator A.

## 1. Background

Circulatory system diseases (CSDs) are characterised by high mortality, hospitalisation rates and heavy medical burden. They have become one of the most serious diseases and adversely affect human health. According to Chinese government’s health statistics, heart diseases and cerebrovascular diseases (both CSDs) rank as the top two causes of morbidity and mortality rates amongst rural residents in China [1]. The hospitalisation rates of patients with CSDs in rural China have increased by 3.29-fold to 20.4% in 2013 from 6.2% in 2003 [2,3]. Researchers have attributed this rapid increase to inappropriate admissions [4,5]. The term “inappropriate admission” refers to patients receiving unnecessary hospitalisation services rather than outpatient services [6,7,8]. Inappropriate admission is a major form of excessive demand for inpatient services [9]. It will increase the economic burden of patients and waste medical resources [10,11].

The high rate of inappropriate admissions in county hospitals is related to the local healthcare system of rural China. First, although the Chinese government launched the family physician policy in 2016 to guarantee that most rural residents will have a contracted first-visit general practitioner (GP), most GPs have limited medical ability and play minor roles as opinion leaders in the decision-making process for seeking healthcare. Thus, patients decide whether they will seek healthcare. The high autonomy of healthcare choice [8] results in high hospitalisation rates. Secondly, the New Rural Cooperative Medical System (NRCMS) in China focuses on hospitalisation reimbursements and provides individuals with improved access to inpatient care. Thus, the NRCMS acts as a policy incentive for residents to select hospitalisation services [12]. Thirdly, distance is an important factor of high inappropriate admission rates given the extensive geography of rural areas. Residents are highly likely to choose hospitalisation services to avoid undergoing several rounds of outpatient services when considering the commute between the hospital and their homes [5].

The unclear boundary between outpatient treatment and hospitalisation for CSD cases [13] increases the uncertainties of decision-making by doctors. It also results in the higher hospitalisation rate for CSDs than that for common diseases and may also lead to inappropriate admissions. A study in 1977 showed that the rate of the inappropriate admission of patients with acute ischemic heart disease to Boston Hospital in the United States was as high as 50% [14]. In 2013, the avoidable admission rate of congestive heart failure in 30 hospitals in Chinese cities was 13.01% [15], and the inappropriate admission rate of CSD cases in county hospitals in five midwestern regions in China reached 17.5% [8]. The hospitalisation rates for urinary and digestive system diseases, injury and poisoning are lower than those for CSDs and are approximately 14.0%, 13.3% and 7.0%, respectively [8]. 

The inappropriate admission of patients with CSDs occurs worldwide and poses a challenge to national health management. The provision of oversight and the control of inappropriate hospitalisation by using technical methods remain major concerns. The Appropriateness Evaluation Protocol (AEP) is the main and most effective technique used to evaluate inappropriate admission worldwide [16,17]. It can be applied universally in all disease systems and considers diseases severity and requirement for services as indicators [5]. Most countries use AEP indicators without distinguishing amongst specific diseases, but The Netherlands [18] evaluates severity of illness according to disease category. The application of AEP to the otolaryngology department in a local hospital in Germany has provided poor results because AEP indicators lack disease categorisation. The construction of a special AEP for the otolaryngology department in accordance with disease characteristics has been proposed [19]. Some studies in China have shown that the current AEP indicators for different diseases have different utilisation rates [20].

Global AEP or indicators specific for CSDs do not exist in current studies. However, we found two most active specific indicators of CSDs in international AEP practice, which are: (1) abnormal blood pressure; (2) acute ischemic electrocardiogram report, suspicion of acute myocardial infarction. Both indicators focus on diseases severity [21]. These indicators are crucial for recognising the inappropriate admission of patients with CSDs. We hypothesised that active indicators of CSD cases are different given the different policies and environments of rural areas of China. Identifying specific active indicators is the first step in determining the appropriate inpatient service and controlling the inappropriate admissions of patients with CSDs.

This study aims to analyse the distribution of CSD indicators in order to determine key indicators from appropriate inpatient cases in rural China, and to provide actual suggestions for developing a specific AEP for CSDs in further research. 

## 2. Methods 

### 2.1. Setting and Participants

We conducted this study in western and middle rural China in 2017. We excluded the eastern region of China given its developed economy to eliminate the impact of regional economic level on inappropriate admission rates. We selected Dingyuan County, Anhui Province; Huining County, Gansu Province; Weiyuan County, Gansu Province; and Yilong County, Sichuan Province via stratified random sampling.

The characteristics of the main medical insurance policies of NRCMS in these four counties were similar and included reimbursement and payment. The reimbursement rate of inpatients ranged from 70% to 85%, and the deductible line ranged from 450 RMB to 650 RMB. Outpatient service in county hospitals was not covered. Overall, NRCMS hospitalisation benefits were high in all regions. We selected the largest county-level hospital in each county as the survey area.

### 2.2. Sampling and Data Collection

Rural patients who were hospitalised with CSDs were selected as subjects. The International Classification of Diseases 10th revision (ICD-10) was adopted to code CSDs. The coding range was I00 to I99. Hypertension, ischemic heart disease, heart failure and cerebrovascular disease were specifically included.

The calculation of sample size in previous studies has shown that the estimated inappropriate admission rate P of patients with CSDs is approximately 18% [8], the expected error δ is 0.028, and the confidence level α is 0.05.
$$N=Z_{\alpha /2}^{2}\times P\left(1-P\right)/\delta^{2}=1.96^{2}\times 0.18\times \left(1-0.18\right)/0.028^{2}=724$$

Based on the population in the surveyed area and the operation scale of the surveyed hospital, 200 medical records of inpatient CSD cases in Dingyuan County, Anhui Province and Yilong County, Sichuan Province were selected via mechanical random sampling., and 220 inpatient medical records of CSDs in Weiyuan and Huining of Gansu Province. A total of 794 valid records were ultimately obtained after the removal of non-evaluable medical records.

### 2.3. Measurement

We collected the medical records of inpatient CDS cases from sample hospitals. Information, including personal, hospitalisation and medical treatment information, were extracted from the medical records. Personal information included name, gender, age and health status. Hospitalization information included admission diagnosis, admission and discharge dates and disease severity during admission. The treatment information included medical examinations and results, physical signs and results, surgical status, treatment status and other clinical service information. 

### 2.4. Medical Record Evaluation

The evaluation was performed by special trained personnel, all of whom were professionals in health policy research, had received Ph.D. degrees and were committed to making fair judgments of records in a strictly standardised manner. The improved rural appropriateness evaluation protocol (RAEP), which was based on AEP criteria and was developed by our team in 2013 [5], was used to identify each indicator and to evaluate the appropriateness of admission. 

The RAEP comprises nine requirements for services (Indicator A) and 21 indicators for disease severity (Indicator B) (see Appendix A). Indicator A is expressed as “A1, A2...A9”, and Indicator B is expressed as “B1, B2...B21” [5]. Firstly, experts evaluated each item of inpatient CSD cases. A case was positive and considered as an appropriate admission when the case information conformed to any indicator. The experts then used RAEP to evaluate 794 CSD cases. A total of 529 patients with CSDs, 66.62% of the cases, were judged to have been appropriately admitted. 

### 2.5. Statistics Analysis

The medical record information of inpatients was encoded by using Epidata 3.2 (The EpiData Association, Odense, Denmark). The frequency and proportion of positive indicators, single positive items (only one indicator is active in one case) and multipositive items (two or more indicators are active in one case) were analysed by using IBM SPSS Statistics 22.0 (IBM, Armonk, NY, USA) to identify interactions, especially interaction between Indicators A and B. 

Three-level logistic regression analysis was used to determine the relationship between the independent variables of sociodemographic characteristics (age, gender, year, region and disease) and positive indicators. Dependent variables were positive indicator items (single positive item of Indicator A = 0, single positive item of Indicator B = 1 and common positive item of Indicator A and B = 2). The regression model is as follows:$$LogitP=ln\left(\frac{P}{1}-P\right)=\beta_{0}+\beta_{1}Age_{ij}+\beta_{2}\;Gender_{ij}+\beta_{3}Year_{ij}+\beta_{4}Region_{ij}+\beta_{5}Disease_{ij}+\varepsilon \;$$
where P denotes the probability of the positive status of the indicators, $\beta_{0}$ is the constant term of the model, ε represents the uncontrollable random error term.

### 2.6. Research Ethics

The research methods and investigation tools adopted in this study were approved by the Ethics Committee of Tongji Medical College, Huazhong University of Science and Technology (IORG No: IORG0003571). Written informed consent was obtained from each county hospital. Patient information was anonymised and identified prior to the analysis.

## 3. Result

### 3.1. Characteristics of Patients Who were Appropriately Admitted to County Hospitals

The overall prevalence of appropriate admissions in the four counties was 66.6% and varied amongst counties. Weiyuan had the highest rate of inappropriate admissions (43.7%), whereas Yilong had the lowest rate (24.6%). The rates of appropriate admission between sexes and amongst disease categories showed no differences. Inappropriate admissions were mainly found for patients aged 26 years old to 65 years old (Table 1). 

### 3.2. Total Positive Frequency of Each Indicator

The positive frequencies of Indicator A were higher than those of Indicator B (Figure 1 and Figure 2). A2 (varying dosage/drug under supervision) showed the highest rate of 58.22%, followed by A8 (stopping/continuing oxygen inhalation, 38.19%), A7 (Electrocardiogram per 2 hours, 34.22%) and A3 (calculate intake/output volume, 31.19%). 

B14 (abnormal blood condition), B3 (severe in electrolyte/blood and vigour) and B12 (abnormal blood pressure) were the top three positive items of Indicator B with frequencies of 27.98%, 13.61% and 9.26% respectively. B10 (dehiscence of surgical wound), B18 (spinal cord lesions) and B21 (burns on specific areas) were not found in sample cases. A2, A3, A8, A7 and B14 were the top five positive indicators amongst all CSD cases.

### 3.3. Positive Frequencies of Single or Multiple Indicators 

We found that indicators were highly likely to show multipositives. Only 118 cases (22.31%) were identified as appropriate admissions by only one indicator (Figure 3 and Figure 4). A2, A3, A8 and B14 were the most active single positive indicators (Table 2).

As to multipositive, 2–3 indicators combined to identify appropriate admission of CSDs was the main form, and it account for 65.97% (Table 2). A total of 62 cases (11.72%) had positive frequencies for more than four indicators. A2 had the highest frequency (20.23%) amongst the two multipositive indicators. The frequency of A2 was approximately 2.23 times higher than that of a single positive frequency (Table 3). A7 only showed 2 cases for single positive.

### 3.4. Comparison of the Frequencies of Indicators A and B

The frequency of a single positive item for Indicator A, which is mainly used to measure the requirement for inpatient services, was higher (17.58%) than that for Indicator B (4.73%). Meanwhile, Indicator A remained dominant (73 cases, 13.80%) even in two-indicator-positive form. A total of 58 cases (10.96%) were positive for Indicators A and B, whereas only four cases were positive for Indicator B (0.76%). 

### 3.5. Analysis of the Influencing Factors of Positive Indicators

Three-level logistic regression was used to analyse the factors that influenced the distribution of positive indicators. The dependent variables were the positive indicators. Single positivity for A (224 cases, 42.30%), single positivity for B (33 cases, 6.20%) and common positivity for A and B (272 cases, 51.40%) were used as reference categories respectively for three regressions, and the results of the same conclusion were combined. ‘Women’, ‘Yilong’ and ‘Other Diseases’ were considered as the reference categories of their corresponding independent variables. The logistic regression results showed that the four variables of gender, year, region and disease type are the influencing factors of positive indicators (see Table 4).

The regression results and the comparison of multipositivity for Indicators A and B indicate that males were more likely to be single positive for indicator A than females (odds ratio (OR) = 0.612). Single positivity for indicator A (OR = 0.642) and Multipositivity for A and B (OR = 1.408) increased with years. Patients in Huining were more inclined to be single positive for Indicator B than those in Yilong (OR = 3.455; OR = 0.297). Coronary heart disease was more inclined to be single positive for Indicator A (OR = 0.563) and hypertension III was more inclined to be single positive for Indicator B (OR = 0.278) than other diseases. 

## 4. Discussion

### 4.1. Inappropriate Admission Rates of Circulatory System Disease (CSD) Cases Vary Across Different Regions

We found that the inappropriate admission rate of CSD cases reached 33.38% and vary in three sampling counties. Meanwhile, the rate is higher than that found in previous studies [5,8], It was due to the different NRCMS policy in sampling time and areas. CSDs mainly occur amongst the elderly [22,23]. The inappropriate admission rates for the elderly over 60 years of age in rural China have reached 23.2% [24]. The high mortality rate and the unclear boundary of inpatient and outpatient services [13] also stimulate the rapid increase in the inappropriate admission rate of CSD cases.

### 4.2. Indicators with High-Positive Rates in CSD Cases 

A2, A3, A8, A7 and B14 are the top five positive indicators amongst the 30 indicators. Firstly, the high positive rate for A2 is attributed to the need of drug adjustment and treatment for patients with CSDs under daily supervision by doctors to ensure the rationality and safety of medications [25]. Secondly, the calculation of the input and the output of body fluids for CSD inpatients provides important guidance for doctors to make medical decisions [26,27] and cannot be completed in outpatient treatments. This finding accounts for the positive frequency of A3. Thirdly, A8 is one of the top five positive indicators given by which inpatients with CSD usually present ischemia and hypoxia symptoms, so oxygen therapy is required to ensure blood oxygenation [28], which can not be accessed in rural residents’ home in rural China. Fourthly, B14, a treatment measure indicator, reflects severe blood abnormalities with serious consequences, such as microcirculation disturbance, blood stasis and thrombosis. B14 is an important disease indicator [29]. Patients with CSD require inpatient hospitalisation treatments. A7 is more likely to be multipositive than other indicators. It represents an indispensable service for CSD cases, and patients require hospitalisation for real-time ECG monitoring or indication measurements [30]. 

### 4.3. Difference between the High Rural Appropriateness Evaluation Protocol (RAEP) Positive Rate Index in this Study and the International AEP of CSDs

The two most active specific indicators of CSDs in international AEP are B12 (abnormal blood pressure) and B13 (ventricular fibrillation/acute myocardial ischemia) in RAEP. However, we found that only B12 was relatively active. Indicator A (requirement for services) was a more sensitive single and multipositive indicator of CSD cases in rural China than Indicator B (disease severity). This finding was further validated by the outcomes of three-level logistic regression, where Indicator B often runs with Indicator A. Our results attributed to the different policies and environments of rural China. These differences, in turn, stimulate the increase in inappropriate admission rate. 

### 4.4. Indicator Characteristics of Patients Who Were Appropriately Hospitalised for CSDs

The results of three-level logistic regression suggest that the influence of gender, year, region and disease type on the positive status of indicators should be emphasised when judging the admission of patients with CSDs. Males tended to access inpatient services even when only one positive indicator item, particularly Indicator A, was found because they are the main source of family labour and have high social position [31]. By contrast, females seek medical treatment for illness after considerable delay and then access inpatient service. Thus, Indicator B was positive for females. However, the number of residents accepting hospitalisation increased with time even if only Indicator A was positive. Patients with coronary heart disease are prone to sudden myocardial infarction, and the main invasive treatment for elderly patients with coronary heart disease is coronary artery surgery [32]. Hypertension III is more serious than hypertension, and its clinical manifestations are severe [33]. It is highly consistent with Indicator B, which represents the severity of illness. Therefore, the single positive item of Indicator B is more in hypertension III.

## 5. Conclusions

Our analysis of the distribution of the indicators of appropriate CSD admission revealed that the key indicators A2, A8, A7, A3 and B14 are helpful for evaluating the appropriateness of CSD case admission. Indicator A, whether alone or in combination with Indicator B, was more active than Indicator B. The results of regression analysis showed that the role of Indicator B became limited over time. We recommend that managers focus on these indicators, particularly Indicator A, to improve the efficiency and accuracy of supervision. 

## 6. Limitation

This study has the following shortcomings: Firstly, the sample area was limited and only included county-level hospitals. However, rural residents often go to city and township hospitals in China. Secondly, our sample had a size of approximately 800 and thus cannot represent the entire status of China. Thirdly, although we analysed the distribution of each indicator, we did not investigate the actual effect of each positive indicator in CSD cases. Therefore, in future studies, we can delve into the specific impact of each positive indicator on CSDs. Similarly, the indicator distribution and the influence of specific positive indicators can also be studied for other diseases with high inappropriate admission rate. 

## Figures and Tables

**Figure 1 ijerph-16-01621-f001:**
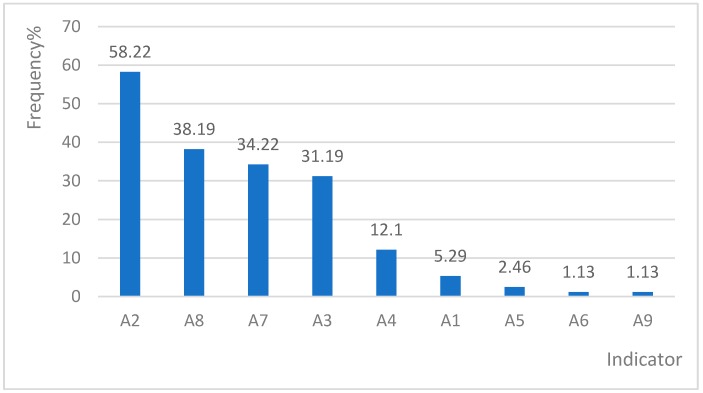
Positive frequency of indicator A in the sample area.

**Figure 2 ijerph-16-01621-f002:**
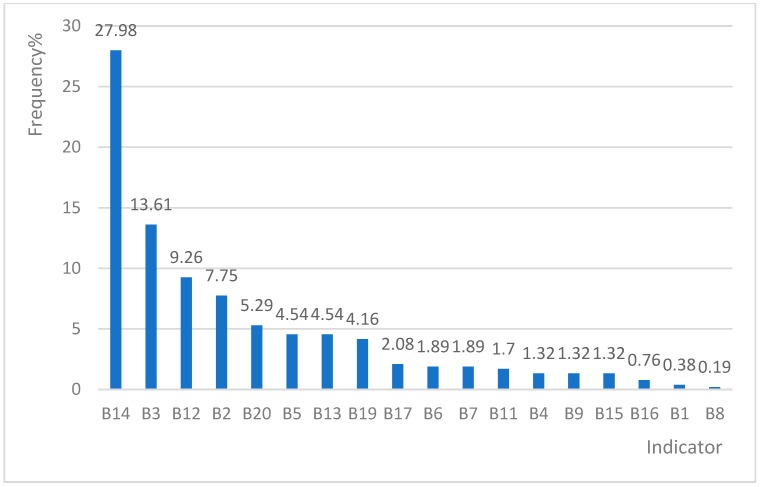
Positive frequency of indicator B in the sample area.

**Figure 3 ijerph-16-01621-f003:**
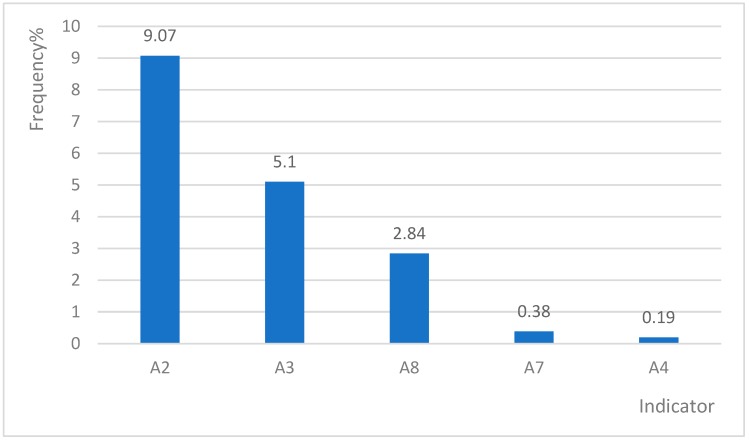
Single positive frequency of indicator A in the sample area.

**Figure 4 ijerph-16-01621-f004:**
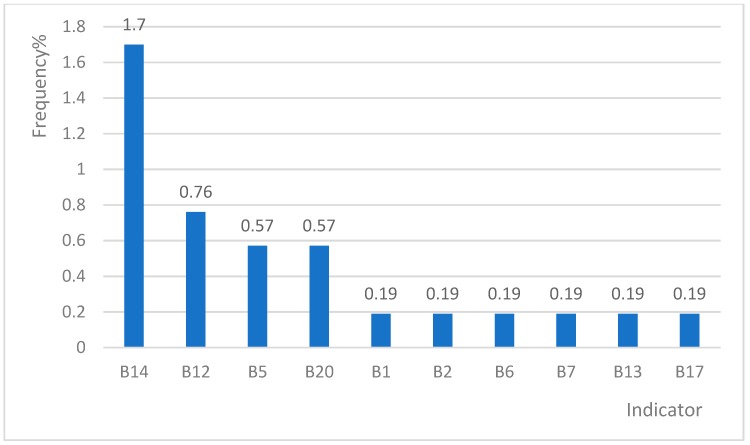
Single positive frequency of indicator B in the sample area.

**Table 1 ijerph-16-01621-t001:** Distribution of cases and appropriate admission (*n* = 794).

Variable	All (Column %)	Appropriateness of Admission		*p*-Value
		Yes Number (Line %)	No Number (Line %)	
All	794	529 (66.6)	265 (33.4)	
County				
Weiyuan	215 (27.1)	121 (56.3)	94 (43.7)	0.001
Dingyuan	193 (24.3)	132 (68.4)	61 (31.6)	
Huining	219 (27.6)	150 (68.5)	69 (31.5)	
Yilong	167 (21.0)	126 (75.4)	41 (24.6)	
Gender				
Male	384 (48.4)	257 (66.9)	127 (33.1)	0.764
Female	399 (50.3)	263 (65.9)	136 (34.1)	
Age, years				
Less than 25	9 (1.1)	9 (100.0)	0 (0.0)	0.008
26–45	44 (5.5)	28 (63.6)	16 (36.4)	
46–65	309 (38.9)	186 (60.2)	123 (39.8)	
More than 65	426 (53.7)	302 (70.9)	124 (29.1)	
Mean (SD)	65.32 (12.867)			
Disease category				
Cerebral infarction	137 (17.3)	85 (62.0)	52 (38.0)	0.255
Coronary heart disease	117 (14.7)	79 (67.5)	38 (32.5)	
hypertension	50 (6.3)	32 (64.0)	18 (36.0)	
hypertension III	53 (6.7)	30 (56.6)	23 (43.4)	
Other	437 (55.0)	303 (69.3)	134 (30.7)	

**Table 2 ijerph-16-01621-t002:** Positive frequency of the main indicator.

Indicator	Total Positive	Single Positive	Multi Positive of 2 Indicators	Multi Positive of 3 Indicators	Multi Positive of 4 Indicators
A2	308 (58.22%)	48 (9.07%)	107 (20.23%)	69 (13.04%)	39 (7.37%)
A8	202 (38.19%)	15 (2.84%)	39 (7.37%)	54 (10.21%)	40 (7.56%)
A7	181 (34.22%)	2 (0.38%)	41 (7.75%)	59 (11.15%)	33 (6.24%)
A3	165 (31.19%)	27 (5.10%)	46 (8.70%)	37 (6.99%)	26 (4.91%)
B14	148 (27.98%)	9 (1.70%)	45 (8.51%)	29 (5.48%)	29 (5.48%)

**Table 3 ijerph-16-01621-t003:** Multi-positive frequencies of the indicators in the sample area.

Multi-positive of 2 Indicators			Multi-positive of 3 Indicators			Multi-positive of 4 Indicators		
Indicator	Number of Case (*n*)	Proportion (%)	Indicator	Number of Case (*n*)	Proportion (%)	Indicator	Number of Case (*n*)	Proportion (%)
A2 + A7	33	6.24	A2 + A7 + A8	18	3.40	A2 + A7 + A8 + B14	9	1.70
A2 + B14	28	5.29	A3 + A4 + A8	8	1.51	A3 + A4 + A8 + B3	6	1.13
A2 + A8	16	3.02	A2 + A8 + B14	6	1.13	A2 + A7 + A8 + B13	3	0.57
A8 + B14	9	1.70	A2 + A3 + A7	5	0.95	A3 + B2 + B3 + B14	3	0.57
A3 + B3	8	1.51	A2 + B3 + B14	5	0.95	A1 + A2 + A3 + A7	2	0.38
A2 + B12	7	1.32	A7 + A8 + B14	5	0.95	A2 + A3 + A7 + B14	2	0.38
A7 + A8	7	1.32	A3 + A4 + B3	4	0.76	A3 + A7 + A8 + B14	2	0.38
A2 + A3	6	1.13	A2 + A7 + B12	3	0.57	--	--	--
A3 + A8	6	1.13	A2 + A7 + B14	3	0.57	--	--	--
A3 + B14	6	1.13	A2 + A7 + B20	3	0.57	--	--	--
A3 + A4	5	0.95	A2 + B12 + B14	3	0.57	--	--	--
Others	38	7.18	Others	54	10.21	--	--	--
Total	169	31.95	Total	117	22.12	Total	63	11.91

**Table 4 ijerph-16-01621-t004:** Multivariate logistic regression analysis of the positive indicators of patients with CSDs (*n* = 529).

Indicator	B vs A		C vs A		C vs B	
	COR(95%CI)	AOR(95%CI)	COR(95%CI)	AOR(95%CI)	COR(95%CI)	AOR(95%CI)
Age	0.993(0.966 to 1.020)	1.002(0.973 to 1.033)	0.998(0.984 to 1.011)	0.999(0.985 to 1.013)	1.005(0.979 to 1.032)	0.997(0.967 to 1.027)
Gender						
Male	0.442(0.207 to 0.944)*	0.514(0.229 to 1.153)	0.642(0.448 to 0.919)*	0.612 (0.422 to 0.887)**	1.450(0.686 to 3.068)	1.190(0.536 to 2.645)
Female (Ref)						
Year	0.565(0.411 to 0.777)*	0.642 (0.457 to 0.903)*	0.904(0.769 to 1.062)	0.904(0.758 to 1.079)	1.600(1.171 to 2.186)**	1.408 (1.009 to 1.965)*
Region						
Weiyuan	3.375(1.020 to 11.166)*	2.646(0.768 to 9.122)	1.009(0.600 to 1.698)	0.979(0.566 to 1.695)	0.299(0.0902 to 0.976)*	0.370(0.109 to 1.253)
Dingyuan	0.233(0.025 to 2.148)	0.323(0.034 to 3.119)	0.999(0.608 to 1.642)	1.127(0.650 to 1.956)	4.294(0.468 to 39.382)	3.486(0.365 to 33.255)
Huining	3.375(1.064 to 10.701)*	3.455 (1.044 to 11.437)*	0.869(0.531 to 1.421)	1.027(0.610 to 1.730)	0.257(0.082 to 0.809)*	0.297 (0.091 to 0.968)*
Yilong (Ref)						
Disease type						
Cerebral infarction	1.131(0.352 to 3.638)	1.646 (0.479 to 5.656)	1.642(0.975 to 2.767)	1.601(0.922 to 2.779)	1.452(0.468 to 4.501)	0.973(0.296 to 3.193)
Coronary heart disease	0.501(0.140 to 1.793)	0.707 (0.185 to 2.696)	0.565(0.338 to 0.942)*	0.563 (0.327 to 0.968) *	1.126(0.312 to 4.070)	0.796(0.207 to 3.057)
hypertension	1.379(0.363 to 5.229)	0.823 (0.204 to 3.327)	0.631(0.293 to 1.360)	0.563(0.256 to 1.237)	0.458(0.118 to 1.767)	0.684(0.169 to 2.778)
hypertension III	3.394(1.139 to 10.114)*	2.260 (0.649 to 7.869)	0.657(0.285 to 1.516)	0.628(0.262 to 1.508)	0.194(0.064 to 0.589)**	0.278 (0.080 to 0.966)*
Other diseases (Ref)						

A = single positive of Indicator A, B = single positive of Indicator B, C = common positive of A and B. B vs. A: with A as the reference category, the independent variable tends to B. C vs. A: with A as the reference category, the independent variable tends to C. C vs. B: with B as the reference category, the independent variable tends to C. Ref: reference category. * *P* < 0.05, ** *P* < 0.01.

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
