# Peer review of "Distribution of the Indicator of the Appropriate Admission of Patients with Circulatory System Diseases to County Hospitals in Rural China: A Cross-Sectional Study"

_ijerph, 2019, doi:10.3390/ijerph16091621_

Round 1

Reviewer 1 Report

Introduction

: overall clear, with an adequate reference to the relevant literature. Just the initial and the final parts are a little bit confusing. Please, re-write those parts.

Discussion

: many sentences are difficult to read. Please, revise it carefully. Furthermore, some of the conclusions stated appear to be quite speculative, therefore needing to be more related to literature findings. Future developments are missing and should be added.

Author Response

Dear Reviewer 1,

Thanks for your kind commending for this paper and your valuable suggestions. Our responses to your comments are below:

Comment 1: Introduction: overall clear, with an adequate reference to the relevant literature. Just the initial and the final parts are a little bit confusing. Please, re-write those parts.

Response: It is very good suggestions and we agree it is necessary. We have carefully read and re-write those parts.

Page 1-2 Line 41-50: Circulatory System Diseases (CSDs) are characterised by high mortality, hospitalisation rates and heavy medical burden. They have become one of the most serious diseases and adversely affect human health. According to Chinese government’s health statistics, heart diseases and cerebrovascular diseases (both belong to CSDs) rank as the top two causes of morbidity and mortality rates amongst rural residents in China [1]. The hospitalisation rates of patients with CSDs in rural China have increased by 3.29-fold to 20.4% in 2013 from 6.2% in 2003 [2,3]. Researchers have attributed this rapid increase to inappropriate admissions [4,5]. The term “inappropriate admission” refers to patients receiving unnecessary hospitalisation services rather than outpatient services [6,7,8]. Inappropriate admission is a major form of excessive demand for inpatient services [9]. It will increase the economic burden of patients and waste medical resources [10,11].

Page 2-3 Line 85-96: Global AEP or indicators specific for CSDs do not exist in current studies. However, we found two most active specific indicators of CSDs in international AEP practice, they are: 1) abnormal blood pressure; 2) acute ischemic electrocardiogram report, suspicion of acute myocardial infarction. Both indicators focus on diseases severity [21]. These indicators are crucial for recognising the inappropriate admission of patients with CSDs. We hypothesised that active indicators of CSD cases are different given the different policies and environments of rural areas of China. Identifying specific active indicators is the first step in determining the appropriate inpatient service and controlling the inappropriate admissions of patients with CSDs.

This study aims to analyse the distribution of CSD indicators, to determine key indicators from appropriate inpatient cases in rural China, and to provide actual suggestions for developing a specific AEP for CSDs in further research.

Comment 2: Discussion: many sentences are difficult to read. Please, revise it carefully. Furthermore, some of the conclusions stated appear to be quite speculative, therefore needing to be more related to literature findings. Future developments are missing and should be added.

Response: Thanks for your valuable suggestions. We have reread and revised the discussion part carefully. We checked all the conclusions and made sure that the relevant literature findings was supported. We added the future developments in the limitation part in Page 12 Line 287-290.

Page 11-12 Line 229-274:

4.1. Inappropriate admission rates of CSD cases vary across different regions

We found that the inappropriate admission rate of CSD cases reached 33.38% and vary in three sampling counties. Meanwhile, the rate is higher than that found in previous studies [5,8], It was due to the different NRCMS policy in sampling time and areas. CSDs mainly occur amongst the elderly [22,23]. The inappropriate admission rates for the elderly over 60 years of age in rural China have reached 23.2% [24]. The high mortality rate and the unclear boundary of inpatient and outpatient services [13] also stimulate the rapid increase in the inappropriate admission rate of CSD cases.

4.2. Indicators with high-positive rates in CSD cases

A2, A3, A8, A7 and B14 are the top five positive indicators amongst the 30 indicators. Firstly, the high positive rate for A2 attributed to the need of drug adjustment and treatment for patients with CSDs under daily supervision by doctors to ensure the rationality and safety of medications [25]. Secondly, the calculation of the input and the output of body fluids for CSD inpatients provides important guidance for doctors to make medical decisions [26,27] and cannot be completed in outpatient treatments. This finding accounts for the positive frequency of A3. Thirdly, A8 is one of the top five positive indicators given that inpatients with CSD usually present ischemia and hypoxia symptoms , so oxygen therapy is required to ensure blood oxygenation,[28], which can not be accessed in rural residents’ home in rural China. Fourthly, B14, a treatment measure indicator, reflects severe blood abnormalities with serious consequences, such as microcirculation disturbance, blood stasis and thrombosis. B14 is an important disease indicator [29]. Patients with CSD require inpatient hospitalisation treatments. A7 is more likely to be multipositive than other indicators. It represents an indispensable service for CSD cases, and patients require hospitalisation for real-time ECG monitoring or indication measurements [30].

4.3. Difference between the high RAEP positive rate index in this study and the international AEP of CSDs

The two most active specific indicators of CSDs in international AEP are B12 (abnormal blood pressure) and B13 (ventricular fibrillation/acute myocardial ischemia) in RAEP. However, we found that only B12 was relatively active. Indicator A (requirement for services) was a more sensitive single and multipositive indicator of CSD cases in rural China than Indicator B (disease severity). This finding was further validated by the outcomes of three-level logistic regression, where Indicator B often run with Indicator A. Our results  attributed to the different policies and environments of rural China. These differences, in turn, stimulate the increase in inappropriate admission rate.

4.4. Indicator characteristics of patients who were appropriately hospitalised for CSDs

The results of three-level logistic regression suggest that the influence of gender, year, region and disease type on the positive status of indicators should be emphasised when judging the admission of patients with CSDs. Males tended to access inpatient services even when only one positive indicator item, particularly Indicator A, was found because they are the main source of family labour and have high social position [31]. By contrast, females seek medical treatment for illness after considerable delay and then access inpatient service. Thus, Indicator B was positive for females. However, the number of residents accepting hospitalisation increased with time even if only Indicator A was positive. Patients with coronary heart disease are prone to sudden myocardial infarction, and the main invasive treatment for elderly patients with coronary heart disease is coronary artery surgery [32]. Hypertension III is more serious than hypertension, and its clinical manifestations are severe [33]. It is highly consistent with Indicator B, which represents the severity of illness. Therefore, the single positive item of Indicator B is more in hypertension III.

Page 12 Line 287-290: Therefore, in future studies, we can delve into the specific impact of each positive indicator on CSDs. Similarly, the indicator distribution and the influence of specific positive indicators can also be studied for other diseases with high inappropriate admission rate. 

Reviewer 2 Report

Relevant problem, good structure, clear logic flow.

Some minor issues:

Categories A and B appear to be mixed in the abstract. Please check. A few words explaining the rationale for the A - B categories would be helpful.

Line charts are used in the Figures. Usually line charts denote trends. Here they show categories, therefore bar charts would be easier to the reader.

Author Response

Dear Reviewer 2,

Thanks for your kind commending for this paper and your valuable suggestions. We are pleased to response to them point by point and changes in the article. Our responses to your comments are as follows:

Comment 1: Categories A and B appear to be mixed in the abstract. Please check. A few words explaining the rationale for the A - B categories would be helpful.

Response: Thanks for your valuable suggestions, it is a very important question. We apologize that we have mixed the A-B categories. We revised and explain this in Page 1 Line 21-22: RAEP has two parts: Indicator A, which represents requirement for services, and Indicator B, which represents diseases severity.

Comment 2: Line charts are used in the Figures. Usually line charts denote trends. Here they show categories, therefore bar charts would be easier to the reader.

Response: Thanks for your valuable suggestions. We have changed four figures from line chart to bar chart in Page 5-7 Line 179-182,187-190
